# Improving Lurasidone Hydrochloride’s Solubility and Stability by Higher-Order Complex Formation with Hydroxypropyl-β-cyclodextrin

**DOI:** 10.3390/pharmaceutics15010232

**Published:** 2023-01-10

**Authors:** María Elena Gamboa-Arancibia, Nelson Caro, Alexander Gamboa, Javier Octavio Morales, Jorge Enrique González Casanova, Diana Marcela Rojas Gómez, Sebastián Miranda-Rojas

**Affiliations:** 1Facultad de Química y Biología, Universidad de Santiago de Chile, Av. Libertador Bernardo O’Higgins 3363, Estación Central, Santiago 9170022, Chile; 2Centro de Investigación Austral Biotech, Facultad de Ciencias, Universidad Santo Tomas, Avenida Ejército 146, Santiago 8370003, Chile; 3Department of Pharmaceutical Science and Technology, School of Chemical and Pharmaceutical Sciences, University of Chile, Santiago 8380494, Chile; 4Advanced Center for Chronic Diseases, Santiago 8380494, Chile; 5Center of New Drugs for Hypertension, Santiago 8380494, Chile; 6Facultad de Ciencias de la Salud, Instituto de Ciencias Biomédicas, Universidad Autónoma de Chile, Santiago 8910060, Chile; 7Escuela de Nutrición y Dietética, Facultad de Medicina, Universidad Andrés Bello, Santiago 8370321, Chile; 8Departamento de Ciencias Químicas, Facultad de Ciencias Exactas, Universidad Andrés Bello, Av. República 275, Santiago 8370146, Chile

**Keywords:** lurasidone, cyclodextrin, host–guest inclusion, poorly water-soluble drugs, stability

## Abstract

The biopharmaceutical classification system groups low-solubility drugs into two groups: II and IV, with high and low permeability, respectively. Most of the new drugs developed for common pathologies present solubility issues. This is the case of lurasidone hydrochloride—a drug used for the treatment of schizophrenia and bipolar depression. Likewise, the stability problems of some drugs limit the possibility of preparing them in liquid pharmaceutical forms where hydrolysis and oxidation reactions can be favored. Lurasidone hydrochloride presents the isoindole-1,3-dione ring, which is highly susceptible to alkaline hydrolysis, and the benzisothiazole ring, which is susceptible to a lesser extent to oxidation. Herein, we propose to study the increase in the solubility and stability of lurasidone hydrochloride by the formation of higher-order inclusion complexes with hydroxypropyl-β-cyclodextrin. Several stoichiometric relationships were studied at between 0.5 and 3 hydroxypropyl-β-cyclodextrin molecules per drug molecule. The obtained products were characterized, and their solubility and stability were assessed. According to the obtained results, the formation of inclusion complexes dramatically increased the solubility of the drug, and this increased with the increase in the inclusion ratio. This was associated with the loss of crystalline state of the drug, which was in an amorphous state according to infrared spectroscopy, calorimetry, and X-ray analysis. This was also correlated with the stabilization of lurasidone by the cyclodextrin inhibiting its recrystallization. Phase solubility,^1^H-NMR, and docking computational characterization suggested that the main stoichiometric ratio was 1:1; however, we cannot rule out a 1:2 ratio, where a second cyclodextrin molecule could bind through the isoindole-1,3-dione ring, improving its stability as well. Finally, we can conclude that the formation of higher-order inclusion complexes of lurasidone with hydroxypropyl-β-cyclodextrin is a successful strategy to increase the solubility and stability of the drug.

## 1. Introduction

Lurasidone hydrochloride (LRD) is an atypical antipsychotic drug approved by the FDA for treating schizophrenia and bipolar disorder (manic depression) in adults and adolescents [1]. It has been reported to be capable of stabilizing schizophrenia after a short-term usage (~6 weeks) by clinical studies in adult patients using a placebo-controlled, randomized, and double-blinded scheme [2]. It has also shown anxiolytic and antidepressant effects through the combined blockade of the dopamine type-2 (D2) and serotonin type 2A (5-HT_2A_) receptors [3]. It presents low extrapyramidal activation and neuromotor detriment, along with low central nervous system depression, thus showing few adverse effects. In addition, it has minimal muscarinic and α1 receptor inactivation through blockade [2]. This profile suggests a minor propensity towards weight gain and metabolic dysfunction, while it has high potential for improving cognitive deficits related to schizophrenia [4]. 

The chemical name of LRD is [3aR,4S,7R,7aS]-2-[(1R,2R)-2-[4-(1,2-benzisothiazol-3-yl)piperazin-1yl-methyl]cyclohexylmethyl]-hexahydro-4,7-methano-2H-isoindole-1,3-dione hydrochloride (Figure 1), and it belongs to the second generation of the antipsychotic family [5]. The benzisothiazole ring is responsible for inhibiting activity over the D2 and 5-HT2A receptors by hydrogen bonding and aromatic interaction [6]. Another pharmacophore corresponds to a positive ionizable piperazine group and 2H-isoindole-1,3-dione by hydrogen-bonding acceptor groups [6]. This last ring is prone to alkaline hydrolysis, which is the main mechanism of the drug’s degradation. Moreover, due to the presence of sulfur in the benzothiazole ring, LRD undergoes oxidation to form sulfone and sulfoxide. However, the oxidative mechanism is less relevant than alkaline hydrolysis and requires more extreme conditions, such as 10% hydrogen peroxide. LRD was found to be stable under thermal stress, neutral hydrolysis, acid hydrolysis, and photolytic stress [7].

The drug presents polymorphism and has been described in several crystalline forms and as an amorphous solid [8,9]. This phenomenon has huge impact on the bioavailability, processability, and stability of solid pharmaceutical forms [10]. LRD’s pharmaceutical presentations include tablets of 20, 40, 80, and 120 mg, where 40 and 80 mg per day are the recommended doses [1]. LRD’s absorption is between 9 and 19%, and its T_max_ ranges are between 1.5 and 3 h after a single and double dose, respectively [11]. It is eliminated through the feces and urine, accounting for 80 and 9.0%, respectively [12]. While its BCS class has not been reported, it has been suggested that LRD could be a class II drug due to its low aqueous solubility and its high partition coefficient [13]. 

There are several procedures to increase drug solubility, among which the most commonly used for class II and IV drugs are particle size reduction [14], preparation of solid dispersions [15], crystalline state modification [16], and formation of inclusion complexes with cyclodextrins [17], among others. The cyclodextrins enable the production of the drugs in different pharmaceutical forms, including solids and liquids [18]. The latter case is attractive for LRD, because liquids can be delivered by using a syringe, measuring cup, or spoon for dosing adjustment (20 mg to 160 mg) for different patients (e.g., pediatric, adolescents, or adults), diagnoses (e.g., schizophrenia or bipolar I disorder), or situations such as renal or hepatic impairment [19], thereby facilitating its administration.

As a drug delivery system, cyclodextrins offer several advantages: They can be included in liquid, semi-solid, and solid systems [20,21]; these can control drug release [22]. They can be adapted to different routes of administration, e.g., oral, parenteral, ocular, nasal, dermal, oromucosal, etc. [23]; these can be included in modern drug delivery systems, such as microparticles, liposomes, nanoparticles, etc. [24]. Smart hybrid systems can be developed that are capable of improving the disposition of drugs in the therapeutic target by various mechanisms (e.g., pH, temperature, redox, enzymes, etc.) [25,26].

Cyclodextrins are cyclic oligosaccharides formed by 6–8 glucopyranose units, bonded by an α-1–4 glycosidic bond [27]. This macrocycle adopts a spatial disposition that provides a hydrophobic cavity and a hydrophilic outer surface responsible for its aqueous solubility [18]. The primary and secondary hydroxyls on the large and small edges of the macrocycle are able to favorably interact with water molecules. However, the main feature of cyclodextrins is given by the non-polar character of the central cavity, which enables binding of hydrophobic molecules such as drugs, where the water displacement is thermodynamically favored because of their low interaction strength with the inside of the macrocycle [28]. Normally, the nature of the host–guest interactions involves charge transfer, electrostatic interaction, dispersion forces (Van der Waals), hydrogen bonds, and conformational release of tension [29,30,31]. Another interesting property of the host–guest chemistry of cyclodextrins includes their ability to form complexes of different ratios [32]. These properties have attracted the attention of the pharmaceutical industry, where they have been incorporated as excipients [33]. Their main advantages are their high stability under light, heat, and oxidation, their biocompatibility, the reduction in irritant effects on tissues caused by some drugs, and the decrease in flavor and smell, improving the organoleptic properties of drugs.

Among the types of cyclodextrins, beta-cyclodextrin (BCD) is the most widely used because its size allows it to properly host several sizes of molecules, together with its lack of toxicity when orally administered. However, it has low gastrointestinal absorption compared to other cyclodextrin derivatives, and it may cause renal toxicity when it is administered by the parenteral route [17]. Hydroxypropyl-β-cyclodextrin (HPBCD), obtained by the hydroxylation of BCD, is a derivative that has shown lower toxicity [34] and higher solubility [35], making it an excellent candidate to improve the solubility of lipophilic drugs, for which it is necessary to administer high doses in order to achieve a pharmacological response. 

For example, insoluble itraconazole (BCS II class) was formulated by forming a complex with an HPBCD, thereby obtaining the oral liquid medication known as SPORANOX^®^ [17,36]. In the case of LRD, a patent describes the preparation of complexes with HPBCD or sulfobutylether-β-cyclodextrin for oral or injection formulations by spray-drying and lyophilization methods; however, there is a lack of information supporting the encapsulation process, such as physicochemical characterization and loading [37]. Londhe et al. [38] prepared BCD–LRD complexes by kneaded solid dispersion and spray-dried solid dispersion with 1:1 host–guest ratios. Other strategies employed to improve LRD’s solubility include nanosuspensions [13], co-amorphous complexes [39,40,41], hydrotrope formation [42], and nanoemulsions [43]. 

Herein, we present the formation of a higher-order complex between HPBCD and LRD by the slurry complexation method, by which we improved the solubility and stability of the drug. This is accompanied by a complete experimental characterization of the host–guest interactions. 

## 2. Materials and Methods

### 2.1. Materials

Hydroxypropyl-β-cyclodextrin (Kleptose HPB oral grade) with a molar substitution of 0.6 and beta-cyclodextrin (Kleptose) were obtained from Roquette (Lestrem, France). Lurasidone hydrochloride was obtained from Eurofarm Ltd. (Hong Kong, China) and fully characterized. Methanol, acetonitrile, triethylamine, citric acid, dibasic sodium phosphate, and phosphoric acid were purchased from Merck (Darmstadt, Germany). Purified and Milli-Q^®^ water were produced in the laboratory. 

### 2.2. Drug Characterization

LRD was characterized by using several techniques, including ^1^H-NMR in deuterated dimethyl sulfoxide (Bruker Advance 400, Bremen, Germany), direct infusion in an electrospray (ESI-MS) mass spectrometer (Quattro Premier, Micromass Waters, Milford, MA, USA), KBr dispersion-infrared spectroscopy (Spectrum 400, PerkinElmer, Beaconsfield, UK), powder X-ray diffraction (Bruker D8 Advance, Bremen, Germany), particle size distribution (Mastersizer 2000, Malvern, UK), and HPLC analysis (Alliance 2695/ PDA 996 detector, Waters, Milford, MA, USA) with the following conditions: column symmetry C8 150 mm × 3.9 mm, 5 µm (Waters, Milford, MA, USA), mobile phase acetonitrile:triethylamine 0.1% (adjusted to pH 6 with H_3_PO_4_) 7:3, flow 1 mL/min, injection volume of 10 µL, and UV detection at 317 nm. This wavelength was selected based on its absorption spectrum, which presents a maximum at 317 nm associated with the benzothiazole chromophore (Appendix A).

### 2.3. Preparation of Complexes

The procedure was based on the slurry complexation method [32]. Briefly, the appropriate quantities of LRD and HPBCD equivalent to molar ratios of 1:0.5, 1:1, 1:2, and 1:3 were weighed separately. Each mix was deposited in a mortar and mixed gently. Then, 1 mL of cosolvent consisting of methanol:water 1:1 was added for the first mix and subsequently dispersed to form a slurry. Volumes of 2, 4, and 6 mL were incorporated in the other ratios, respectively. The product obtained was dried for 6 h in a natural convection oven (Binder Inc., Bohemia, NY, USA) at 40 °C and then stored in a desiccator until its use. The humidity of the complexes was determined by using a moisture analyzer (HB43-S, Mettler Toledo, Greifensee, Switzerland).

### 2.4. Phase Solubility Analysis

Solutions of HPBCD ranged from 10 to 300 mg/mL and were prepared in water. LRD was added to each glass vial with HPBCD solution to obtain saturated solutions, including one without HPBCD. These samples were incubated in a water bath dual-action shaker (PolyScience, Inc., Warrington, PA, USA) for 24 h at 100 cycles/minute and 37 °C. Aliquots were filtered with a PVDF 0.45 µm syringe filter (Merck Millipore, Darmstadt, Germany) and analyzed by UV at 314 nm using a UV–Vis Spectrophotometer (Agilent 8453, Santa Clara, CA, USA). The same procedure was carried out with aqueous BCD solutions ranging from 1.1 to 17 mg/mL. 

### 2.5. Infrared Spectroscopy Analysis of Complexes

Approximately 10 mg of each sample was thoroughly ground with KBr in a mortar and a pellet was formed using a manual press. Infrared spectroscopy measurements were obtained in the range of 400–4000 cm^−1^ using a Spectrum 400 (PerkinElmer, UK) with 10 scans by spectra. 

### 2.6. X-ray Diffraction Analysis of Complexes

The crystallinity of the complexes was evaluated with a Bruker D8 Advance instrument (Bruker, Bremen, Germany) using Cu K-alpha radiation (λ = 1.5406 Å, 40 kV, 30 mA). Diffractions were measured in the angular range of 2–80° (2θ).

### 2.7. Differential Scanning Calorimetry of Complexes

Differential scanning calorimetry (DSC) was performed using a DSC131 device (SETARAM Inc., Cranbury, NJ, USA) to determine the physical state of the components in the inclusion complexes. Each sample (10 mg) was placed in an aluminum pan and heated at 10 °C/min from 25 °C to 350 °C.

### 2.8. NMR Measurements of Complexes

The proton nuclear magnetic resonance (NMR) spectrum of each sample was recorded on a Bruker Avance 400 spectrometer, ^1^H, 400.13 MHz (Bruker, Bremen, Germany), in deuterated dimethyl sulfoxide at a temperature of 300 K. Chemical shifts (in ppm) for ^1^H were reported relative to Me_4_Si (TMS).

### 2.9. Efficiency of Encapsulation and Drug Loading

The quantitation of the free drug was performed to determine the efficiency of encapsulation (EE) with the following equation: $$EE\;\left(\%\right)=\frac{\left[Theoretical\;drug\;added\;\left(mg\right)-free\;drug\;\left(mg\right)\right]}{Theoretical\;drug\;added\;\left(mg\right)}\times 100\%$$

For this, about 10 mg of each complex was accurately weighed and transferred into a glass vial with cap. The, 5 mL of chloroform was added, the vial was capped, and the suspension was mixed for 60 min to dissolve any free LRD. Then, the sample was filtered with a hydrophobic PTFE 0.45 µm syringe filter (Merck Millipore, Darmstadt, Germany), and the supernatant was recovered in a tube. The solvent was removed in a nitrogen stream, and the sample was reconstituted with 10 mL of cosolvent acetonitrile:water 1:1, which was then injected into the HPLC according to the conditions described in Section 2.2. 

Drug loading (DL) was determined by LRD direct assay of the complexes with the following equation: $$DL\;\left(\%\right)=\frac{LRD\;assayed\;in\;the\;sample\;\left(mg\right)}{Weight\;of\;the\;sample\;\left(mg\right)}\times 100\%$$

In this case, about 10 mg of each complex was accurately weighed and transferred into a glass vial with cap. Then, 10 mL of cosolvent acetonitrile:water 1:1 was added, and the vial was capped and mixed for 60 min. Then, the sample was filtered with a PVDF 0.45 µm syringe filter (Merck Millipore, Darmstadt, Germany) and analyzed by HPLC according to the conditions described in Section 2.2. 

### 2.10. Solubility of Complexes

A saturated solution of each complex was prepared in citrate–phosphate buffer (McIlvaine buffer) at pH 3.5 and was incubated in a water bath dual-action shaker (PolyScience, Inc., Warrington, PA, USA) for 24 h at 100 cycles/minute and 37 °C. Aliquots were filtered with a PVDF 0.45 µm syringe filter (Merck Millipore, Darmstadt, Germany) and analyzed by HPLC according to the conditions described in Section 2.2, with the exception of changing the mobile phase to acetronitrile:triethylamine 0.1% (adjusted pH 3 with H_3_PO_4_) 35:65 and UV detection at 230 nm, due to peak splitting. 

### 2.11. Dissolution Test of Complexes

An amount of each complex equivalent to 20 mg of LRD was accurately weighed and manually encapsulated in gelatin capsules of size 1. Then, the dissolution testing was carried out in an apparatus II (paddle method) using a Distek dissolution system model 2500i (Distek, Inc., North Brunswick Township, NJ, USA) with 900 mL of citrate–phosphate buffer pH 3.8 (37 °C) and the paddle rotating at 50 rpm, according to the FDA’s recommendations [44]. Each capsule was added to the vessel with a sinker, and aliquots were withdrawn at 5, 10, 15, 30, 45, and 60 min of the test. Each sample was filtered with a PVDF 0.45 µm syringe filter (Merck Millipore, Darmstadt, Germany) and analyzed by HPLC according to the conditions described in Section 2.9.

### 2.12. Evaluation of the Stability-Indicating Method

The method was validated as recommended by the ICH [45], determining all of the analytical parameters with the exception of robustness. To evaluate the specificity, the samples were subjected to stress by temperature (80 °C—2 days), acid hydrolysis (HCl 0.1 N—2 h/40 °C), alkaline hydrolysis (NaOH 0.1N—2 h /room T°; NaOH 0.1N—2 h/40 °C and NaOH 1.0 N—2 h/room T°), and oxidation (H_2_O_2_ 15%—2 h/room T°; H_2_O_2_ 1.5%—2 h/40 °C and H_2_O_2_ 15%—24 h/40 °C). The statistical parameters were calculated using the data analysis tool of the Excel program (Microsoft Corporation, Redmond, WA, USA), and the signal–noise ratio was calculated using the Empower™ version 2 software (Waters, USA).

### 2.13. Degradation Stress Test

Alkaline forced degradation was carried out with LRD and the complexes. An equivalent amount of 10 mg of LRD was added to a glass vial with cap. Then, 1 mL of NaOH 0.1 N was added, and the vial was capped and shaken for 120 min at 80 °C. At the end, each sample was neutralized with 1 mL of HCl, completed to 10 mL with solvent mix (ACN:Water 1:1), filtered with a PVDF 0.45 µm syringe filter (Merck Millipore, Darmstadt, Germany), and analyzed by HPLC according to the conditions described in Section 2.2. For oxidative stress, an amount equivalent to 10 mg of LRD was added to a glass vial with cap, 1 mL of H_2_O_2_ 15% *v*/*v* was added, and the vial was capped and shaken for 24 h at 40 °C. At the end, each sample was completed to 10 mL with solvent mix (ACN:Water 1:1), filtered with a PVDF 0.45 µm syringe filter (Merck Millipore, Germany), and analyzed by HPLC according to the conditions described in Section 2.2.

### 2.14. Stability of Complexes

Accelerated stability of the complexes was assessed in a stability chamber (Memmert, Germany) set to 40 °C and 75% humidity. Each product was placed in a glass dish and stored for three months. Samples were extracted at 1, 2, and 3 months and were analyzed by infrared spectroscopy (Spectrum 400, PerkinElmer, UK), moisture analysis (HB43-S, Mettler Toledo, Switzerland), and the remaining LRD was measured by HPLC according to the conditions described in Section 2.2.

### 2.15. Model building and Docking

The basis for the molecular models for BCD and HPBCD was derived from a previous study carried out by one of the authors, in which these two hosts were complexed with catechin [46]. There, the best-suited HPBCD was selected according to its ability to reproduce NMR experimental observations, and this HPBCD was used in this study. The LRD ligand was built using Avogadro v1.2.0 software [47], and a geometric optimization followed by a conformational search was carried out using the metadynamics approach as implemented in the x-TB v6.5.1 software [48], through the crest module [49,50]. The lowest energy conformation together with the BCD and HPBCD systems was refined through the use of density functional theory (DFT) calculations. For this purpose, the BP86 [51,52] exchange–correlation functional was used because of the compromise between accuracy and efficiency, complemented by the use of the resolution of identity approximation to increase the computational efficiency due to the size of the systems [53,54]. The def2-SVP basis set was used for all of the atoms [55]. All DFT calculations included the dispersion correction developed by Grimme with the damping function to correct for close interactions [56,57,58,59]. The refined geometries were used as the starting coordinates for the docking studies. 

The docking calculations were performed using the AutoDock4 software [60] by means of the Lamarckian genetic algorithm as a search method, as implemented in the program. Each docking run consisted of 100 runs, with the maximum number of evaluations set to 25,000,000 and a maximum number of generations up to 27,000. From the results, we selected the lowest-energy solution to obtain a starting complex. We first proceeded by docking the LRD in its deprotonated form into the BCD and HPBCD hosts, from which we were able to select one final conformation. Then, for the addition of a second cyclodextrin for the LRD–BCD and LRD–HPBCD complexes, the best solution for each complex was docked with a BCD and an HPBCD as a ligand, respectively. Then, the same procedure was followed using LRD in its protonated form (LRDH) for comparison purposes.

### 2.16. Conformational Search and Interaction Energies

The 1:1 and 1:2 complexes obtained from the docking calculations were further refined by a two-stage process. First, we conducted a conformational search using the complexes as starting coordinates by means of the metadynamics approach available on x-TB. We found three low-energy solutions that characterized the three possible binding modes of the deprotonated form of LRD for the following complexes: LRD–BCD, LRD–HPBCD, BCD–LRD–BCD, and HPBCD–LRD–HPBCD. These were optimized by using DFT calculations at the same level of theory used for the refinement of the ligands, namely, the DFT/BP86/def2-SVP methodology. Once the DFT-optimized geometries were obtained, the solvation effects were studied by using the implicit solvation model known as COSMO [61]. Meanwhile, for the protonated form of LRDH, we generally found two distinguishable binding modes with the lowest interaction energies, although only one was found for the HPBCD–LRDH–HPBCD complex.

All interaction energies for the first and second inclusion processes were calculated according to the following expression:Eint = [LRD-cyclodextrin] − [LRD + cyclodextrin]
and for the second inclusion process:Eint = [cyclodextrin-LRD-cyclodextrin] − [LRD-cyclodextrin + cyclodextrin]
where, in order to calculate the interaction energies using the protonated form of the ligand, LRD is replaced by LRDH in the equations. Finally, we calculated the IR spectra of the complex with the lowest interaction energy to compare and properly characterize the experimental results. The interaction energies were decomposed into physical contributions through the second-generation energy decomposition analysis based on absolutely localized molecular orbitals (ALMO-EDA) [62,63,64], using the Q-Chem 5.2 software [65,66]. According to this approach, each interaction energy is decomposed into six terms: ΔEint = ΔE_CT_ + ΔE_POL_ + ΔE_DISP_ + Δ_EELEC_ + ΔE_PAULI_
where ΔE_ELEC_ incorporates the stabilization due to the coulombic electrostatic interactions between the frozen fragment charge distributions. ΔE_POL_ describes the stabilization due to the intramolecular density relaxation caused by the perturbation induced by the presence of the opposite fragment in the complex. ΔE_CT_ is the charge-transfer term that accounts for the energy lowering due to the intermolecular charge transfer between fragments caused by orbital interactions; it also includes the intramolecular density relaxation induced by the charge transfer. ΔE_DISP_ represents the contributions coming from the dispersion forces, which are very important in the host–guest context. Finally, ΔE_PAULI_ corresponds to the electronic repulsion due to the interaction between electrons of the same spin, which can also be interpreted as electronic steric repulsion due to volume-exclusion effects when two fragments are compressed.

## 3. Results

### 3.1. Drug Characterization

LRD does not appear in official compendia such as the United States Pharmacopeia (USP), so it was fully characterized. Table 1 shows a summary of the tests employed, where the chemical structure of the drug was confirmed by proton nuclear magnetic resonance (^1^H-NMR), infrared spectroscopy (IR), and mass analysis (ESI-MS). Interestingly, by ^1^H-NMR, a salt proton at 10.31 (s, 1H), benzothiazole aromatic region from 8.13 to 7.46 (4 H), and isoindole-1,3-dione group at 4.03 ppm (m, 2H) were among those that were confirmed by the ¹H-¹H correlation spectroscopy experiment (Appendix A). In the case of IR, the absorptions associated with N-H stretching (3437 cm^−1^), salt overtone N-H (2262 cm^−1^), and the symmetric/antisymmetric carbonyl in isoindole-1,3-dione (1762–1688 cm^−1^) stand out (Appendix A). The mass spectra presented a peak associated with the parent ion, with a difference of 0.03 Da between the experimental and theoretical results. In addition, there was an isotopic peak at 495.2289 m/z associated with one atom of sulfur (^34^S) in the structure (Appendix A). The drug was a crystalline solid with an X-ray diffraction pattern (Appendix A) conforming to that described by Reddy et al. [67] and micronized with d(90%): 7.308 µm (Appendix A). The API met the general pharmacopeia criteria of purity (98–102%), with 99.88% according to HPLC, and the chromatographic peak was symmetric and spectroscopically clean with an experimental angle less than the threshold of purity (Appendix A). 

### 3.2. Phase Solubility Analysis

Phase solubility studies of LRD were carried out with BCD and HPBCD. The graph obtained for BCD (insert in Figure 2) had a linear relationship (R = 0.99) with an A_L_ type plot according to the Higuchi and Connors classification [68]. A similar result was reported by Londhe et al. [38]; however, we obtained a positive deviation plot (Ap) with a higher concentration of HPBCD (orange line, Figure 2), suggesting a higher-order complex between LRD and HPBCD. The solubility of the drug was above 20 mg/mL with HPBCD (300 mg/mL), which could facilitate the production of a liquid dosage form. As a reference, SPORANOX^®^ oral solution containing 10 mg/mL itraconazole was solubilized with 400 mg/mL HPBCD [17,36,69]. In addition, the stability constant (K_1:1_) of the LRD–HPBCD complex was calculated from the straight-line portion of the phase solubility plot, resulting in a K_1:1_ = 133.2 M^−1^. This value is within the range from 85 to 6800 M^−1^ reported by some drug–HPBCD formulations [70]. Based on these results, complexes with different guest–host ratios were elaborated by the slurry method and characterized. 

### 3.3. Infrared Analysis

The mean absorption bands of HPBCD were 2935 and 1158 cm^−1^, associated with the C-H alkyl backbone and C-O, respectively (blue line, Figure 3). These were observed at 2932 and 1156 cm^−1^ in the LRD–HPBCD (1:3) complex (pink line), respectively, indicating potential changes in the host structure when the drug is included. In the case of LRD, absorptions were observed at 2938, 2262, and 1762–1688 cm^−1^, assigned to the C-H alkyl group, N-H overtone salt, and C=O in the isoindole-1,3-dione ring, respectively. These signals appeared at the same wave numbers in the physical mixture (green line), but in the LRD–HPBCD (1:3) complex the carbonyl group was displaced to 1769–1696 cm^−1^ and the second band disappeared, indicating that the drug lost its crystalline structure when the inclusion complex was formed. This last change was also reported when synthesizing co-amorphous complexes of LRD [40,41]. Previously, for LRD–BCD complexes obtained by spray-drying and kneaded solid dispersion, only in the first method was a similar result observed [38]. In our case, with the slurry method, only the higher ratios presented this phenomenon, with a diminished band in the blue line and absent in the pink line (Figure 4). At the other ratios (red and black lines), the band was present, indicating partial incorporation of the drug in the HPBCD and a remaining crystalline interaction with the hydrochloride salt. The alkyl and carbonyl groups were not displaced. 

### 3.4. X-ray Diffraction Analysis

The diffraction patterns of complexes are shown in Figure 5. At the 1:2 and 1:3 LRD–HPBCD ratios, the spectra did not have significant peaks, indicating the amorphous nature of the solid. This was consistent with the disappearance of the band at 2262 cm^−1^ in the IR analysis shown in Figure 4. However, crystalline signals of the drug were observed in the 1:0.5 and 1:1 LRD–HPBCD complexes, resulting in a partial encapsulation. In another study for BCD–LRD, the kneading method resulted in a crystalline solid, while the spray-drying method resulted in an amorphous product [38]. 

### 3.5. Differential Scanning Calorimetry of Complexes

The thermoanalytical curves of the complexes are shown in Figure 6, and a summary of the main events is presented in Table 2. LRD presented an endo/exothermic effect associated with melting and decomposition of the drug (black line), typical of its crystalline nature, confirming what was previously described [38]. On the other hand, the HPBCD presented a broad endothermic signal (orange line) at the onset temperature of 67 °C (peak maximum 105 °C), associated with the dehydration of the substance [38]. In the physical mixture, the drug’s melting/decomposition were observed at significantly lower temperatures, in accordance with what had already been reported for a physical mixture of BCD with LRD [38]. At the 1:2 and 1:3 ratios of LRD–HPBCD, the DSC curves did not show the endothermic peak, indicating the amorphous nature of the solid. This was consistent with the X-ray analysis shown in Figure 5. The LRD–HPBCD (1:0.5) complex showed a significant endothermic signal at the onset temperature of 227 °C, indicating the presence of the crystalline drug. This phenomenon occurred to a lesser extent in the LRD–HPBC (1:1) complex.

### 3.6. NMR Analysis

Changes in the chemical shifts were observed in different complexes (Figure 7), and the main ^1^H-NMR signals are summarized in Table 3. The salt and alkyl proton (H_16_–H_18_) had upfield shifts of Δδ = −0.678, −0.082, and −0.010 from LRD to LRD–HPBCD (1:3), indicating a cycloalkyl moiety interaction between the drug and the host. In the isoindole-1,3-dione group, downfield shifts were observed in the H_26–_H_30_ protons with Δδ = 0.017, suggesting the incorporation of the ring in the cavity. Aromatic protons (H_6_–H_7_–H_8_–H_9_) had the fewest downfield shifts, with Δδ = 0.005 from LRD to LRD–HPBCD (1:3), indicating a weak interaction with HPBCD. All of these results suggest the interaction of the drug with more than one host molecule and the formation of a higher-order complex [71,72]. For BCD, the formation of a 1:1 LRD–BCD complex through aromatic interactions with the benzothiazole ring was reported [38].

### 3.7. EE, DL, Humidity, and Solubility of Complexes

Incomplete encapsulation of the drug was obtained with an LRD–HPBCD ratio of 1:0.5 according to Table 4, with an EE of 76.5%. This means that the incorporation of two drug molecules into the HPBCD cavity is unlikely. On the other hand, encapsulation was improved at the 1:1 ratio (94.4%) and was almost complete at the 1:3 ratio (98.2%). These values could indicate the formation of a higher-order complex that would be complementary to the results obtained by ^1^H-NMR. Drug loading (DL) values ranged from 9.5% to 41.4% as the host–guest ratio was reduced. Considering the preparation of a solid dosage form, a ratio up to 1:2 would be acceptable in terms of the final weight of the drug product required for the desired dose loading [70]. 

In addition, the humidity of the complexes was less than 5%, reaching maximum values in the complexes with ratios of 1:2 and 1:3. The drying procedure was not extended due to a constant value at 24 h. A drug solubility of 0.349 mg/mL (20 °C) in McIlvaine buffer pH 3.5 has been described in the FDA application of the Latuda drug product from Sunovion Pharmaceuticals [73]. Herein, we obtained a solubility ranging from 1.3 to 20 mg/mL as the host–guest ratio was increased. Interestingly, there was a significant change when the ratio was increased from 1:1 (1.9 mg/mL) to 1:2 (18.9 mg/mL). Increases in solubility of approximately 12 mg/mL with an LRD–puerarin complex [40], 15 mg/mL with an LRD nanosuspension [13], and 42 mg/mL with an LRD–cysteine complex [41] have been reported. In two cases [40,41], the formation of a co-amorphous system enhanced the solubility of LRD through intermolecular interactions with the complexing agent. A similar mechanism could be postulated for LRD–HPBCD based on the results obtained here from XRD, IR, and ^1^H-NMR analyses. 

### 3.8. Dissolution Testing

FDA dissolution conditions [38] were used to evaluate the gelatin capsule performance of the complexes. The formation of the inclusion complex increased the drug’s dissolution (Figure 8), which reached approximately 99% for the LRD–HPBCD (1:3) complex at 60 min. A similar profile was observed for the 1:2 ratio, with 96% at 60 min. LRD alone only reached approximately 27% at the end of the test. For the 1:0.5 and 1.1 LRD–HPBCD ratios, the release was slower and incomplete, reaching approximately 57% and 90% at 60 min, respectively. This improvement could be due to the amorphization of the drug, and a similar behavior was obtained with gelatin capsules containing an amorphous complex of LRD–BCD prepared by spray-drying, reaching 100% at 60 min [38]. With the kneading method, a 60% dissolution was obtained at 60 min, where the solid complex was crystalline. Another strategy for increasing the LRD dissolution was a nanosuspension that achieved rapid release (>85% in 30 min) in the pH range of 2 to 6.8 [13].

### 3.9. Evaluation of the Stability-Indicating Method

The method to evaluate the stability of the drug was partially validated, and the results of the parameters required to demonstrate its purpose are shown in Table 5. The technique presented linearity from 0.6 µg/mL to 1.0 mg/mL, and the detection limit was 0.6 µg/mL (Appendix A) with a signal–noise ratio of 3. The statistical test of the intercept indicated that it passed through the origin, and the slope test confirmed that the relationship between the concentration and the chromatographic response of the method was linear.

Regarding the evaluation of the specificity of the method, Table 6 presents the results of the drug stress tests. LRD showed significant degradation under alkaline stress (Appendix A), where five degradation products appeared in the moderate condition (0.1 N NaOH/40 °C—2 h), with 11.2% drug reduction. There was no co-elution of peaks from the other analytes, which was confirmed because the peak purity angle (0.071) was below the threshold (0.326). In the extreme condition, only two peaks were observed, and the LRD chromatographic area was reduced to 47.3%. In oxidative stress, only under extreme conditions (H_2_O_2_ 15% 40 °C—24 h) was there a degradation of 3.7%, with the formation of four peaks. The purity of the peaks indicated that there was no co-elution of the degradants. In the other conditions evaluated, there was no degradation, which is consistent with what was previously published with respect to this drug [7].

### 3.10. Degradation Stress Testing

The instability of LRD in solution occurs mainly by two routes [7]: alkaline hydrolysis yields 19.44% degradation, and an oxidative pathway is responsible for 8.40% degradation. Using HPLC coupled with mass spectroscopy, the products obtained were identified, where the formation of a sulfone and a sulfoxide in the benzisothiazole ring occurred by exposure to hydrogen peroxide, and the generation of a carboxylic derivative and a primary amine occurred after partial and complete cleavage of the isoindole-1,3-dione ring, respectively, under the reaction with sodium hydroxide. Taking this into account, the ability of HPBCD to protect the included drug from basic hydrolysis and oxidative degradation was studied. 

Figure 9 presents the results obtained, where 24.9% degradation was observed for LRD (remaining drug = 75.1%), as expected according to the mentioned mechanism. With the increase in the inclusion ratio (LRD–HPBCD), the hydrolysis was progressively reduced, and in the 1:2 and 1:3 complexes there was practically no drug breakdown, which is consistent with the ability of cyclodextrins to stabilize labile molecules [17,74]. For example, a stabilizing effect of HPBCD was demonstrated on the hydrolytic degradation of methyl salicylate in HCl (pH 1.0) at 65 °C [74]. In the case of oxidative stress, the same protective effect of HPBCD was observed even for the 1:0.5 ratio, showing that this instability pathway is less preponderant in relation to alkaline hydrolysis.

### 3.11. Stability Studies 

The infrared spectra of the complexes subjected to the stability tests (Table 7) were similar to those characterized at the beginning of this study, where the intensity of the band at 2262 cm^−1^ in the complexes disappeared with the increase in the HPBCD ratio (Appendix A). The absence of this band in the 1:3 complex indicated that the amorphous nature was maintained after three months of storage. Assaying for LRD revealed that the amount of the drug was reduced by 2.4% after three months in the 1:0.5 complex and 0.4% in the 1:1 complex; however, no degradation was observed (positive deviation) in the 1:2 and 1:3 ratios. The first result is explained by the presence of an amount of free drug (as described in Table 4), which is more susceptible to degradation, as opposed to the included drug, which is protected by HPBCD. On the other hand, the complexes captured humidity from the environment during the stability study, which increased with the HPBCD ratio, where the greatest difference was observed in the LRD–HPBCD (1:3) ratio but did not exceed 9.0%. This is consistent with the hygroscopic nature of HPBCD, where an increase of over 10% has been reported after excipient exposure to an environment with a relative humidity of 75% [35]. The HPBCD hygroscopicity phenomenon has also been reported in tablets [75,76]. Therefore, this factor must be considered when manufacturing the final pharmaceutical form, choosing a suitable low-permeability primary packaging material or using silica gel in the bottle.

### 3.12. Models for Complex Formation and Stability

To obtain a clearer picture of the possible binding modes and stabilities of LRD in this type of host, we proceeded by obtaining complexes using both BCD and HPBCD for comparative purposes, aiming to shed light on the advantages of the use of HPBCD with respect to BCD. The results associated with the interaction energies (ΔE_int_), the interaction energies corrected by solvent (ΔE_int-S_), and the dipolar moments (μ) are listed in Table 8 and Appendix A. For the first inclusion process, we found a clear dependence of the protonation state of LRD and of the type of host. For the first inclusion process, the LRD–HPBCD complex showed an increase of 14.7 kcal/mol in stability with respect to the BCD counterpart. Meanwhile, the transition from the deprotonated to the protonated LRDH involved an increase of 81.5 kcal/mol, indicating that this would be the preferred state for the ligand when binding to the host. The geometries of the inclusion complexes selected within this study are shown in Figure 10, Figure 11, Appendix A and Appendix A. In terms of the binding mode for the first inclusion process, in all cases—namely, BCD, HPBCD, LRD, and LRDH—the benzisothiazol ring was always located outside the cyclodextrine through the larger opening, also known as a secondary rim. According to this, we found three possible variations in the binding mode: the first involved the inclusion of the cyclohexylmethyl with the isoindole-1,3-dione moieties, corresponding to the complex with the lowest interaction energy with LRD and the highest stability; the second involved the inclusion of the piperazin with the cyclohexylmethyl, partially leaving out the aliphatic part of the isoindole-1,3-dione; finally, the third involved the full inclusion of the isoindole-1,3-dione, leaving the rest outside the secondary rim. Meanwhile, the inclusion process of LRDH was well characterized by two binding modes in both BCD and HPBCD. The first, and the most stable, involved the inclusion of the cyclohexylmethyl moiety, leaving both ends of the LRDH molecule outside, whereas the second binding mode was characterized by the inclusion of the isoindole-1,3-dione. The main difference between the complexes formed by BCD and HPBCD was the fact that, with HPBCD, one of the oxygens from the isoindole-1,3-dione from LRD in both protonation states always formed a hydrogen bond with a hydroxyl group from the side chains of HPBCD, which was absent in all complexes with BCD.

For the second inclusion process to reach a stoichiometry of 1:2, there was a clear increase in the interaction energies, which was more marked for the BCD complexes, which for LRD increased by 51.0 kcal/mol after the addition of the second host, while for LRDH the increase was 38.8 kcal/mol. For the case of HPBCD, the addition of the second host to the LRD–HPBCD complex increased the interaction energy by 33.0 kcal/mol, and with LRDH the increase was 22.1 kcal/mol. Despite these observations, in general terms, the total stability denoted by the magnitude of the interaction energy was always higher for the HPBCD complexes. The binding modes were characterized by an extended conformation of LRD and LRDH for the lowest-energy conformations with BCD and HPBCD. In all cases, it seemed that the titratable nitrogen from the piperazin ring was always located at the interphase between the cyclodextrins, in both LRD and LRDH. The differences observed in the interaction energies for the second inclusion process using LRD and BCD were mostly associated with the proper interaction between the two cyclodextrins. While the lowest-energy conformer showed an ordered conformation, BCD-C2 and BCD-C3 from Appendix A showed a more disordered interaction and conformation between the BCDs. When the ligand was protonated, the decrease in the interaction energy was due to an attempt to reach a folded conformation of LRDH inside the BCD–BCD cavity because of an intramolecular hydrogen bond between the isoindole-1,3-dione and the proton from the piperazin ring. However, this conformation is not favored, as it causes a decrease of 53.8 kcal/mol in the interaction energy, indicating that the extended conformation of the ligand is favored for the 1:2 stoichiometry, independent of its protonation state. The same behavior was observed for HPBCD, where a more ordered interaction interphase between the cyclodextrins led to lower interaction energies of LRD and HPBCD–HPBCD in both protonation states, always favoring the extended conformation of the ligand. Interestingly, for both BCD–BCD and HPBCD–HPBCD when in complex with LRD, no hydrogen bonds were formed, opposite to what was observed at a 1:1 stoichiometry with HPBCD. Only for the protonated form (LRDH) did the proton from the piperazin ring form a hydrogen bond with one of the hydroxyl groups from one of the cyclodextrins for both BCD and HPBCD. 

For all of the cases studied here, the incorporation of the solvent effect caused a decrease in the interaction energies. The reason for this behavior comes from the increase in the stabilization of the cyclodextrin ligand, which is higher than the stabilization observed for the complex and, consequently, causes a decrease in the interaction energy. Experimentally, this destabilizing effect would be analogous to the work required for the desolvation process necessary to incorporate the ligand inside the cyclodextrin. From a qualitative point of view, the increase in the dipolar moment (μ) of the complex with respect to the μ from the ligand reflects the increase in its solubility in water. As shown in Table 7, if we consider the μ of LRDH as a reference, which is 8.15 D, we could suggest that the increase in solubility is mainly associated with the first inclusion process. 

To describe the nature of the host–guest interaction, an interaction energy decomposition analysis was carried out, as detailed in the methodology section; the results are shown in Figure 12. According to the stabilizing contributions, it was noticeable that the interaction was mainly dominated by electrostatic effects, whose contribution was always close to 50%, followed by the dispersion interaction contribution, which ranged between 29 and 38% depending on the complex. The charge transfer contribution due to orbital interactions was low, as expected, as there was no formal covalent interaction. Finally, the polarization contribution was the lowest contribution, indicating minor effects on the electronic density redistribution after complex formation in both the cyclodextrins and the ligands from an intramolecular point of view. The destabilizing effects reflected by the Pauli repulsion partially indicated the steric electronic effects between electrons of the same spin, which were high due to the proximity between the ligand and the host, which was higher for the models representing the 1:2 stoichiometry, as expected. Nevertheless, this destabilizing effect was overcompensated by the stabilizing contributions previously mentioned, leading to the formation of complexes of high stability.

To characterize the experimental bands observed though the IR experiments, we carried out the simulation of the normal vibrational modes and the consequent IR spectrum (results listed in Table 9) to understand the qualitative relationships between the changes in intensity and energy of some bands—more specifically, those associated with the symmetric and antisymmetric double C=O stretching from the isoindole-1,3-dione and the N-H bond for the complexes where the ligand was protonated. For this, we focused on the complexes with HPBCD with the deprotonated and protonated ligands. After proper identification of the bands associated with the normal vibrational modes on which we focused our study, the results showed that the energies calculated for the bands of the free LRDH were overestimated; however, after being complexed with HPBCD in its protonated form, the energies became closer to the experimental results. This reflects that the free LRDH does not properly represent the experimental vibrational modes of the same system, as it is in a saline form as LRD-HCl surrounded by images of itself in the experimental matrix. However, after the complex of LRDH with HPBCD was formed, the Cl^−^ anion seemed to be left out of the system, and the energy bands then properly reproduced the experimental behavior. 

Regarding the complexes with the deprotonated ligand, they all showed an overestimation of the energy bands, although the decrease in intensity when increasing the stoichiometry was qualitatively observed, reinforcing that the LRD ligand was in its protonated form. The comparison of the behavior observed when increasing the stoichiometry from 1:1 and 1:2 between the experimental and the calculated energy bands of the LRDH–HPBCD complexes with the protonated ligand revealed semi-quantitative agreement between the experimental results and the calculations, thereby validating the models obtained. Even though in the experimental spectra the changes were only minor, these were correctly represented by the models. For the symmetric stretching, the experimental shift was from 1762.0 cm^−1^ to 1767.0 cm^−1^, while the calculated shift was from 1789.6 cm^−1^ to 1801.1 cm^−1^. For the antisymmetric stretching, the experimental energy bands shifted from 1688.0 cm^−1^ to 1691.0 cm^−1^, while the model shifted from 1713.7 cm^−1^ to 1728.7 cm^−1^. In addition, the same qualitative decrease in the intensity band observed experimentally was reproduced by the models. Regarding N-H, the main change expected for this band was a decrease in its intensity, which was observed in the models, where the intensity of the N-H stretching decreased from 329.7 to 281.5. However, this phenomenon was not observed in the experimental spectra, due to overlapping of the OH groups (stretching band) of the LRD–HPBCD complex.

## 4. Discussion

The formation of inclusion complexes through host–guest supramolecular chemistry depends on the inherent characteristics of the drug to be encapsulated and the host cavity. More than one molecule can enter a cavity if the drug is of sufficiently small molecular weight, as in the case of pyrene [77]. In general, most of the chemical entities can form 1:1 relationships with cyclodextrins; however, the possibility of forming higher-order complexes with two and three guest units has been presented where, in addition to size, the functionalities present in the drugs can influence the encapsulation process [78]. For LRD, the formation of a 1:1 complex with BCD was described, which was interpreted in terms of the phase solubility diagram. Nonetheless, the limited solubility of this cyclodextrin does not allow the observation of other relationships that occur at higher concentrations [79]. According to the NMR analysis, the 1:1 interaction was interpreted to occur through the aromatic ring, with downfield shifts depending on the manufacturing procedure, with Δδ (ppm) values between 0.00050 and 0.0268. HPBCD has an aqueous solubility of >600 mg/mL [17] and, in our case, at high concentrations of this agent, a positive deviation of the phase solubility graph was observed, suggesting the formation of a higher-order complex. A downfield shift was observed in the NMR of the aromatic protons, with a Δδ (ppm) = 0.005 between the complexes and LRD, similar to that described with BCD, as well as in the isoindole-1,3-dione ring with Δδ (ppm) = 0.017. This last difference, which was not observed in the previous study [38], implies the interaction of this group with HPBCD—possibly through hydrogen bonds—due to the presence of the hydroxyl substituent in this cyclodextrin. This was also observed by the shift of the carbonyl infrared bands of that ring from 1762/1688 cm^−1^ to 1769/1696 cm^−1^ in the 1:3 complex. In addition, a significant upward shift was observed from the salt and the alkyl proton, with Δδ = −0.678, −0.082, and −0.010, which was also not described for BCD, reinforcing the hypothesis of the interaction of more than one ring in the formation of the higher-order complex with HPBCD. The computational models obtained confirm the stability of the complex formed between LRD and HPBCD, and also support the possibility of an energetically favored 1:2 ratio. 

The nature of the solid state plays an important role in the solubility of drugs, with crystalline drugs generally having less solubility than those in an amorphous state [10]. Several patents describe the preparation of amorphous LRD [67,80], with slight improvements in its solubility relative to crystalline forms [9]; however, in general, this type of solid presents stability problems. Qian et al. reported an increase in the solubility of a synthesized amorphous LRD relative to the crystalline form; however, in saturated solutions, the initial increase decayed over time, and in 10 h it equaled the crystalline solubility [39]. The same authors developed a complex of saccharin and LRD that showed a greater intrinsic solubility of the drug through the formation of a co-amorphous solid whose supersaturated solutions were maintained for 24 h. The stability study showed that the amorphous form synthesized after two days showed signs of crystallinity, while the saccharin–LRD complex maintained its amorphous nature for 60 days. Charge-assisted hydrogen bonding interaction between the N^+^-H group of LRD and the C=O group in saccharin was identified as being responsible for the improvements in the drug properties. In our case, with 1:2 and 1:3 ratios of LRD–HPBCD, solids with amorphous characteristics were obtained, which would imply that, in addition to aromatic interactions, other supramolecular bonds would be involved in drug stabilization, such as hydrogen bonds between the isoindole-1,3-dione ring and HPBCD, as suggested by NMR and infrared analyses. This made it possible to significantly improve the solubility of the drug, reaching values of up to 20 mg/mL for the complex with a 1:3 ratio.

In addition to the physical changes that an amorphous solid may undergo over time, it is important to evaluate the intrinsic stability of the drug in pre-formulation studies and how different excipients influence it [10]. The main instability of LRD is due to the hydrolysis of the isoindole-1,3-dione ring and in our study we found that encapsulation with HPBCD is an effective method to avoid this degradation route. In other studies, only the physical stability of co-amorphous complexes has been evaluated [40,41], without indicating information on chemical stability, and in both cases the interaction did not involve the isoindole-1,3-dione ring, leaving it susceptible to hydrolytic degradation. With respect to accelerated stability studies, the LRD–puerarin complex was stable for 50 days at 40 °C and 75% relative humidity, with a small decrease in titer that was not significant [40]. In our case, the complexes with 1:2 and 1:3 ratios remained stable for 90 days under the same conditions, and the positive variations could be attributed to product uniformity or analytical deviation. 

Complexes with cyclodextrins have proven to be useful for preparing pharmaceutical solutions for poorly soluble drugs [69]. This represents a challenge because these formulations have several excipients, such as cosolvents, sweeteners, and flavoring agents, which can influence the complexation process. For example, propylene glycol, which is used as a cosolvent at a percentage of 10% *v*/*v*, competes in the formation of the 1:2 itraconazole–HPBCD complex, reducing its affinity [81]. For this API with the same excipient, it has also been shown that the manufacturing method can have an influence, where the separate dissolution procedure increases the solubility of itraconazole more than the classic method (all components together) [82]. In another case, the use of a surfactant (poloxamer 188) also improved the solubility of a classical complex between genistein and HPBCD, through the formation of a ternary mixture [83]. Therefore, it is necessary to evaluate the influence of different excipients and the manufacturing method of the LRD–HPBCD complex to obtain a suitable liquid dosage form.

## 5. Conclusions

Cyclodextrins are excipients that are widely used to improve the solubility of poorly soluble drugs, through the formation of inclusion complexes. Herein, we conclude that the LRD–HPBCD complex significantly increases the solubility of the guest molecule when a 1:2 or 1:3 ratio is used. Higher-order complexes also improve drug stability under alkaline and oxidative degradation and under accelerated conditions. LRD medication requires dose adjustments depending on the type of patient; therefore, the increase in solubility could allow the preparation of an oral liquid pharmaceutical form that fulfills this purpose. Additionally, the preparation of a solid through a simple slurry method allows a formulation with higher dissolution that could improve the low bioavailability of this drug or the formulation of a parenteral powder for injectable reconstitution with improved stability. New compatibility and stability studies will be necessary for this purpose.

## Figures and Tables

**Figure 1 pharmaceutics-15-00232-f001:**
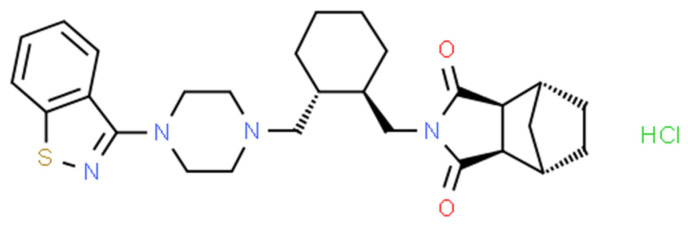
Chemical structure of LRD.

**Figure 2 pharmaceutics-15-00232-f002:**
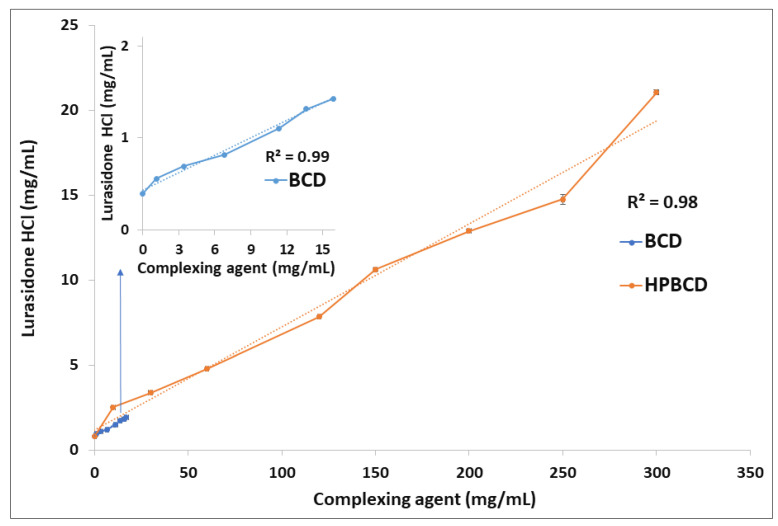
Phase solubility analysis of LRD with BCD (blue line) and HPBCD (orange line).

**Figure 3 pharmaceutics-15-00232-f003:**
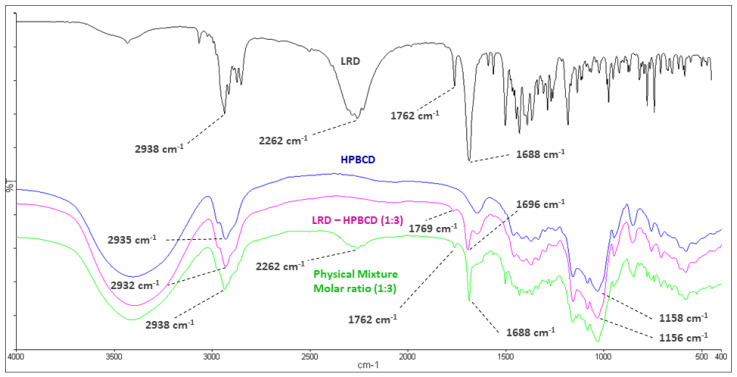
Infrared spectroscopy of LRD (black), HPBCD (blue), physical mixture (green), and LRD–HPBCD (1:3) complex (pink).

**Figure 4 pharmaceutics-15-00232-f004:**
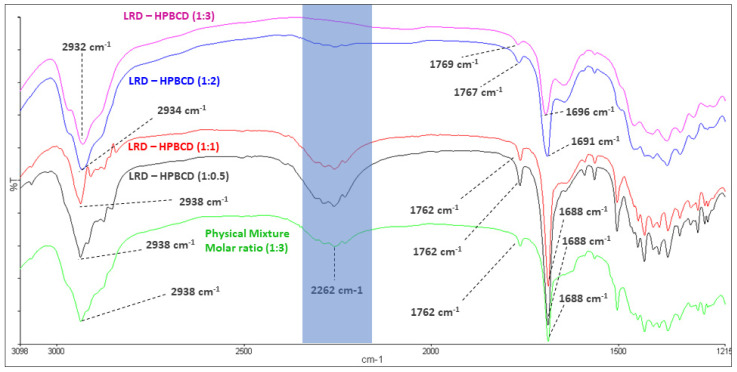
Infrared spectroscopy of LRD–HPBCD complexes.

**Figure 5 pharmaceutics-15-00232-f005:**
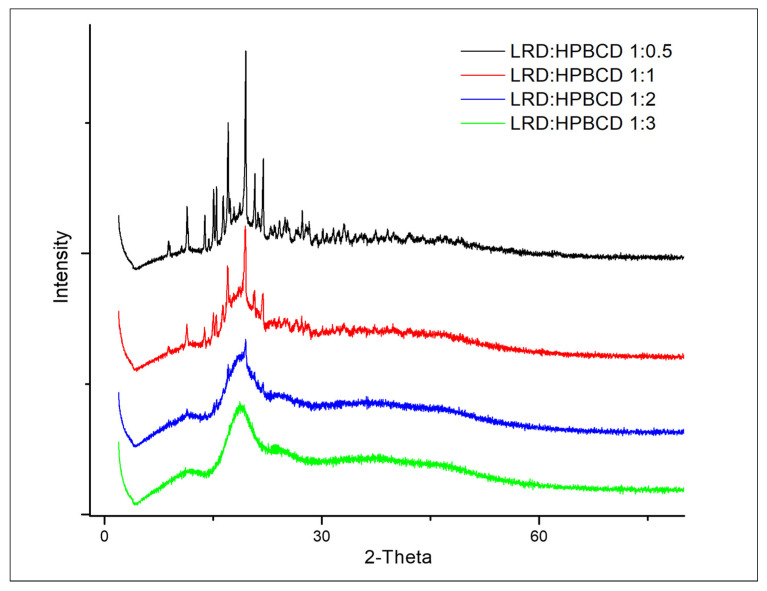
Diffractograms of LRD–HPBCD complexes.

**Figure 6 pharmaceutics-15-00232-f006:**
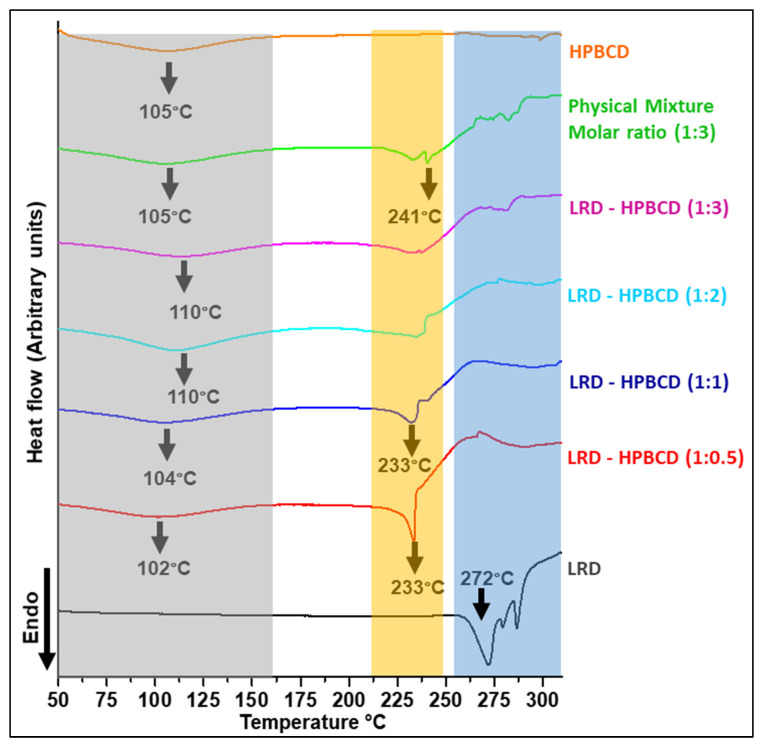
DSC curves of LRD–HPBCD complexes, LRD, HPBC, and PM. The grey band highlights the desolvation, the orange band shows the crystalline transition, and the blue band represents the crystalline phase of the drug.

**Figure 7 pharmaceutics-15-00232-f007:**
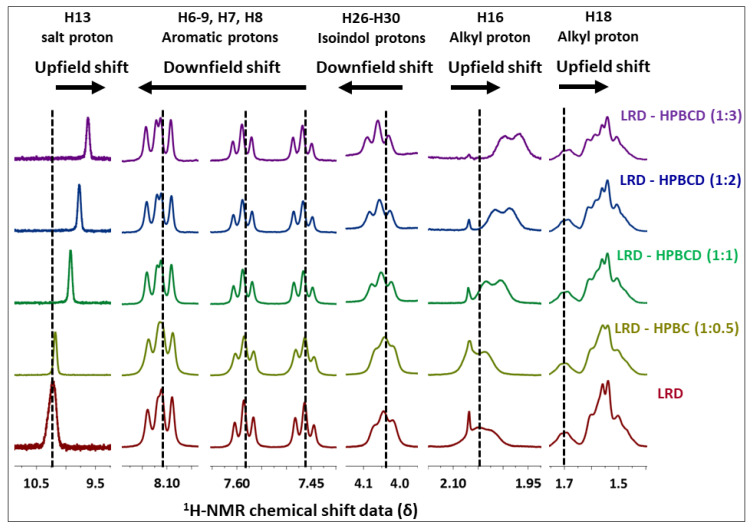
^1^H-NMR of LRD–HPBCD complexes.

**Figure 8 pharmaceutics-15-00232-f008:**
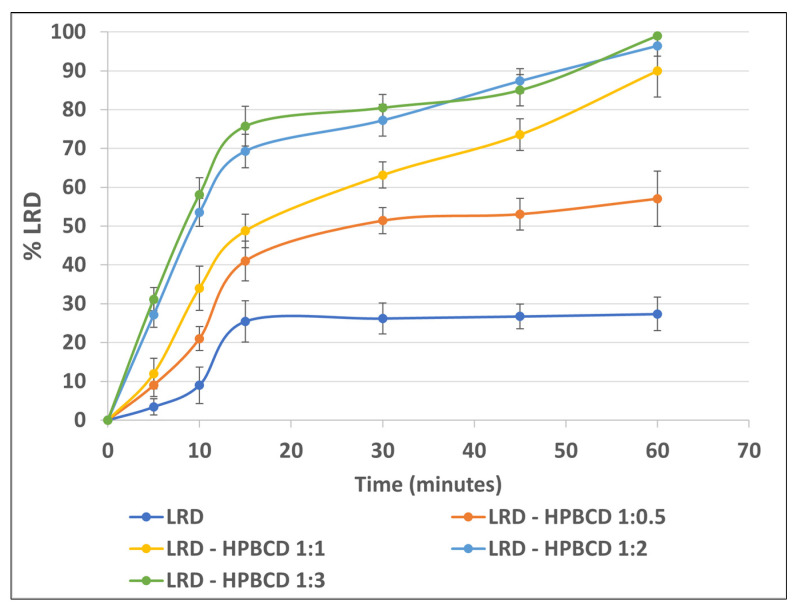
LRD dissolution from the complexes.

**Figure 9 pharmaceutics-15-00232-f009:**
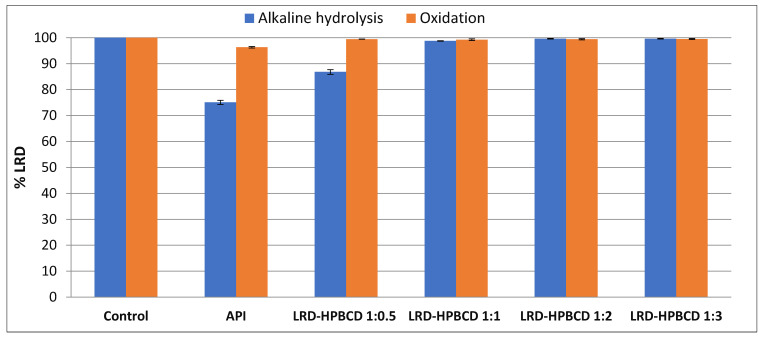
Alkaline and oxidative degradation tests of LRD (API) and complexes.

**Figure 10 pharmaceutics-15-00232-f010:**
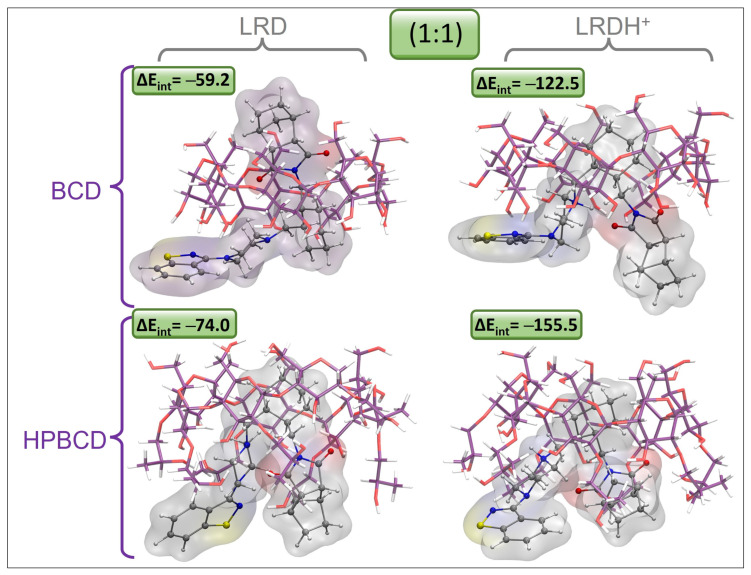
Graphical representation of the geometries of the lowest-energy and most stable inclusion complexes for LRD and LRDH into BCD and HPBCD for the first inclusion process (1:1) (cyclodextrin carbon: purple, LRD/LRDH carbon: gray, hydrogen: white, oxygen: red, nitrogen: blue, sulfur: yellow; ΔE_int_ is the interaction energy in kcal/mol).

**Figure 11 pharmaceutics-15-00232-f011:**
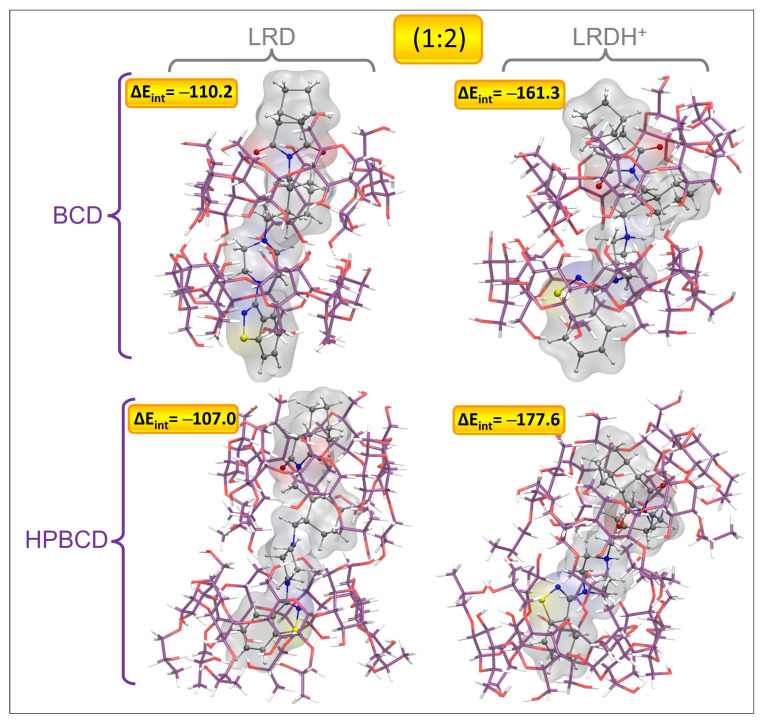
Graphical representation of the geometries of the lowest-energy and most stable inclusion complexes for LRD and LRDH into BCD and HPBCD for the second inclusion process (1:2) (cyclodextrin carbon: purple, LRD/LRDH carbon: gray, hydrogen: white, oxygen: red, nitrogen: blue, sulfur: yellow; ΔE_int_ is the interaction energy in kcal/mol).

**Figure 12 pharmaceutics-15-00232-f012:**
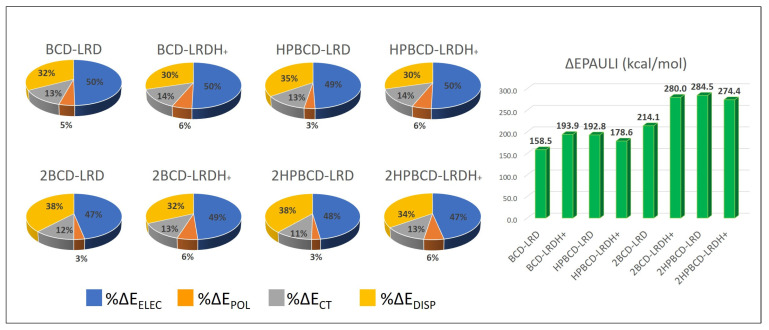
Energy decomposition analysis of the interaction energy for the lowest-energy complexes (most stable conformations). The stabilizing contributions—namely, electrostatic (ΔE_ELEC_), polarization (ΔE_POL_), charge transfer (ΔE_CT_), and dispersion interaction (ΔE_DISP_)—are presented as percentages (%), while the destabilizing contribution (ΔE_PAULI_) is presented in kcal/mol.

**Table 1 pharmaceutics-15-00232-t001:** Analytical characterization of LRD.

Test	Result
^1^H-NMR	Consistent with structure (δ): 10.31 (s, 1H), 8.13 (d, J = 8.5, 1H), 8.10 (d, J = 8.5, 1H), 7.59 (t, J = 7.5, 1H), 7.46 (t, J = 7.5, 1H), 4.03 (m, 2H), 3.56 (m, 6H), 3.27 (m, 4H), 3.08 (m, 1H), 2.68 (m, 2H), 2.49 (m, 1H), 2.05 (m, 1H), 1.69 (m, 1H), 1.53 (m, 6H), 1.30 (m, 2H), 1.15 (m, 4H), and 0.95 (m, 2H).
Infrared spectroscopy	Consistent with structure (cm^−1^): 3437, 3067, 2938, 2262, 1762, 1688, 1591, 1565, and 742.
MS-ESI+	Consistent with structure: ESI+ ion peak = 493.2226 m/z Theoretical ion calculated = 493.2559 m/z (based on monoisotopic mass: 492.2559 Da)
Powder X-ray diffraction	Crystalline solid with mean diffraction peaks at 11.39, 13.86, 15.06, 15.46, 16.39, 17.06, 19.49 (100% relative intensity), 20.74, and 21.90 degrees (2θ angle).
Particle size distribution	d(90%): 7.308 µm, d(50%): 3.518 µm, and d(10%): 1.607 µm
HPLC	Purity 99.88%. Purity angle = 10.570 (purity threshold: 66.174) USP plate count = 7545/USP tailing = 0.96

**Table 2 pharmaceutics-15-00232-t002:** Summary of the main endothermic events of the DSC experiments.

Sample	Onset (°C)	Offset (°C)	Peak Maximum (°C)
HPBCD	67	142	105
PM ^a^	74 239	143 249	105 241
LRD–HPBCD (1:3)	80	151	110
LRD–HPBCD (1:2)	76	157	110
LRD–HPBCD (1:1)	70 222	141 236	104 233
LRD–HPBCD (1:0.5)	63 227	142 234	102 233
LRD	262	276	272

^a^ PM: physical mixture (molar ratio 1:3).

**Table 3 pharmaceutics-15-00232-t003:** Shifted mean ^1^H-NMR protons in the LRD–HPBCD complexes.

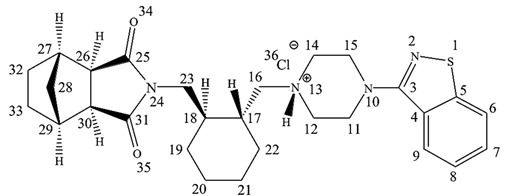						
Assigned Proton	^1^H-NMR Chemical Shift Data (δ) Complexes					Δδ ^a^
	LRD	LRD–HPBCD (1:0.5)	LRD–HPBCD (1:1)	LRD–HPBCD (1:2)	LRD–HPBCD (1:3)	
H-N13 salt	10.306	10.181	9.922	9.774	9.627	−0.678
H6 H9	8.138	8.138	8.141	8.141	8.143	0.005
H7	7.585	7.585	7.588	7.588	7.590	0.005
H8	7.462	7.462	7.466	7.466	7.467	0.005
H26 H30	4.044	4.042	4.051	4.057	4.061	0.017
H16	2.049	2.036	2.005	1.986	1.967	−0.082
H18	1.690	1.693	1.689	1.683	1.680	−0.010

^a^ Δδ = Difference in δ between LRD–HPBCD (1:3) complex and LRD.

**Table 4 pharmaceutics-15-00232-t004:** EE, DL, humidity, and solubility of complexes.

Sample	EE (%)	DL (%)	Humidity (h%)	Solubility (mg/mL) ^a^
LRD–HPBCD (1:0.5)	76.5 ± 0.40	41.4 ± 0.11	3.23 ± 0.13	1.3 ± 0.08
LRD–HPBCD (1:1)	94.4 ± 0.10	23.6 ± 0.11	3.75 ± 0.16	1.9 ± 0.10
LRD–HPBCD (1:2)	96.3 ± 0.20	13.9 ± 0.07	4.40 ± 0.23	18.9 ± 0.60
LRD–HPBCD (1:3)	98.2 ± 0.20	9.5 ± 0.03	3.95 ± 0.18	20.0 ± 0.40

^a^ Solubility in citrate–phosphate buffer (McIlvaine buffer) pH 3.5.

**Table 5 pharmaceutics-15-00232-t005:** Validation summary of the HPLC method.

Parameter	Indicator	Result	Criteria	Complies
Accuracy	Recovery	Mean 100.5%	98–102%	Yes
Precision	RSD ^a^	0.4	RSD: NMT ^b^ 2%	Yes
Linearity	Correlation coefficient R	0.9999 (0.6 µg to 1 mg/mL)	≥0.99	Yes
Linearity	Intercept test (*t*-test)	0.73	t exp ^c^. ≤ t table (2.78)	Yes
Linearity	Slope test (*t*-test)	409	t texp. ≥ t table (2.78)	Yes
Range	Recovery	Mean 101.9%	95–105%	Yes
Range	Correlation coefficient R	0.9999 (5.0 µg to 1 mg/mL)	≥0.99	Yes
Range	RSD	RSD max = 2.5	RSD: NMT 3%	Yes
Quantitation limit	Signal-to-noise (s/n) > 10	s/n = 10 (3.0 µg/mL)	(s/n) ≥ 10	Yes
Detection limit	Signal-to-noise (s/n) > 3	s/n = 3 (0.6 µg/mL	(s/n) ≥ 3	Yes

^a^ RSD: Relative standard deviation; ^b^ NMT: no more than; ^c^ t exp= t Student experimental.

**Table 6 pharmaceutics-15-00232-t006:** Specificity evaluation using the stress test.

Condition	% LRD	% D ^a^	Peaks ^b^	Purity Angle	Purity Threshold	Co-Elution
NaOH 0.1 N room T°—2 h	97.4	2.6	4	0.938	3.477	No
NaOH 1.0 N room T°—2 h	47.3	52.7	2	0.159	0.372	No
NaOH 0.1 N 40 °C—2 h	88.8	11.2	5	0.071	0.326	No
H_2_O_2_ 15% room T°—2 h	99.4	0.6	3	3.693	12.842	No
H_2_O_2_ 1.5% 40 °C—2 h	99.5	0.5	3	4.062	12.634	No
H_2_O_2_ 15% 40 °C—24 h	96.3	3.7	4	4.698	10.482	No
80 °C—48 h	100	0	0	4.958	12.943	No
HCl 0.1 N	100	0	0	4.528	12.811	No
Control	100	0	0	5.078	10.637	No

^a^ % D: % degradation; ^b^ number of degradation peaks.

**Table 7 pharmaceutics-15-00232-t007:** Accelerated stability of the complexes.

Sample	Infrared Analysis (at 3 Months) ^a^	DL (%) (at 3 Months)	ΔDL (%) ^b^	Humidity (H%) (at 3 Months)	ΔH% ^b^
LRD–HPBCD (1:0.5)	Present	39.0 ± 0.17	−2.4	6.20 ± 0.15	+2.97
LRD–HPBCD (1:1)	Present	23.1 ± 0.14	−0.4	7.53 ± 0.09	+3.78
LRD–HPBCD (1:2)	Reduced	14.6 ± 0.07	+0.8	8.76 ± 0.17	+4.36
LRD–HPBCD (1:3)	Absent	9.7 ± 0.15	+0.2	8.75 ± 0.12	+4.80

^a^ Absorption at 2262 cm^−1^; ^b^ difference between the third month and the beginning of study (Table 4).

**Table 8 pharmaceutics-15-00232-t008:** Interaction energies (ΔEint), interaction energy corrected by implicit solvent (ΔEint-S), solvent destabilization contribution (Solv), dispersion interaction energy contribution (Edisp), and dipolar moments (μ) for the complex formation between lurasidone and BCD or HPBCD in a (1:1) and (1:2) stoichiometry. Energies are in kcal/mol and μ is in debye (D).

Complex	ΔE_int_	ΔE_int-S_	Solv %	μ ^a^
(1:1)				
LRD–BCD	−59.3	−42.9	27.7	12.78
LRDH^+^–BCD	−122.5	−95.2	22.3	15.78
LRD–HPBCD	−74.0	−53.6	27.6	9.72
LRDH^+^–HPBCD	−155.5	−126.4	18.7	14.14
(1:2)				
LRD–BCD	−110.2	−80.4	27.0	8.49
LRDH^+^–BCD	−161.3	−144.3	10.5	11.43
LRD–HPBCD	−107.0	−80.4	24.9	15.06
LRDH^+^–HPBCD	−177.6	−152.6	14.1	21.98

^a^ The reference dipolar moments of LRD in its neutral and charged forms are 3.45 D and 8.15 D, respectively.

**Table 9 pharmaceutics-15-00232-t009:** IR simulation frequencies for the symmetric (symm) and antisymmetric (antisymm) normal vibrational modes of the isoindole-1,3-dione moiety and N-H when lurasidone is protonated for the complex formation between lurasidone and HPBCD in a (1:1) and (1:2) stoichiometry. Frequencies are in cm^−1^.

Complex	Symm	Intensity	Antisymm	Intensity	N–H	Intensity
LRD (exp)	1762.0		1688.0		3437.0	
LRD (calc)	1954.1	60.6	1894.5	502.4		
LRDH^+^ (calc)	1966.6	138.6	1869.5	668.1	3434.4	36.0
(1:1)						
LRD–HPBCD (exp)	1762.0		1688.0		OH ^a^	
LRD–HPBCD	1953.4	62.1	1878.8	762.9		
LRDH^+^–HPBCD	1789.6	53.9	1713.7	736.6	2956.8	329.7
(1:2)						
LRD–HPBCD (exp)	1767.0		1691.0		OH ^a^	
LRD–HPBCD	1958.1	33.8	1886.9	740.5		
LRDH^+^–HPBCD	1801.1	32.9	1728.7	658.6	2922.3	281.5

^a^ Band overlapped by OH stretching of the LRD–HPBCD complex.

## Data Availability

Not applicable.

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
