# Peer review of "Improving Lurasidone Hydrochloride’s Solubility and Stability by Higher-Order Complex Formation with Hydroxypropyl-β-cyclodextrin"

_pharmaceutics, 2023, doi:10.3390/pharmaceutics15010232_

Round 1

Reviewer 1 Report

The Authors prepared complexes of lurasidone with hydroxypropyl-beta-cyclodextrin in different molar ratios in order to improve the pharmaceutical behavior of the active principle. They used a range of experimental techniques to perform a through characterization of the complexes obtained.

The poor solubility of the active principles is always a “burning” issue that forces the pharmaceutical industry to face major challenges. Thus, the present manuscript could be of interest for the pharmaceutical researches.

The Authors made a lot of experimental work and used a number of techniques.

The manuscript is quite well written but requires a revision of the English language and some corrections (see, for example, pag. 2, line 66; pag 4, lines 161-162, 182; pag. 6, lines 248, 262; pag. 7 line 290 etc.).

With regard to the FT-IR measurements, the Authors should specify which is the composition of the physical mixture whose spectrum is reported in figure 3. Furthermore, the spectra of all the physical mixtures with the same composition of the complexes should be shown for a comparison  with the spectra of the complexes.

DSC measurements could give important quantitative information starting from the lurasidone melting peak. Can the Authors carry out DSC measurements?

Pag. 14, line 434-435: A compound in the amorphous state has higher solubility than in the crystalline phase. The Authors wrote the opposite. Please, correct.

The paragraph of the Conclusions is poor. The Authors could merge part of the Discussion to the Conclusion paragraph.

Author Response

The Authors prepared complexes of lurasidone with hydroxypropyl-beta-cyclodextrin in different molar ratios in order to improve the pharmaceutical behavior of the active principle. They used a range of experimental techniques to perform a through characterization of the complexes obtained. The poor solubility of the active principles is always a “burning” issue that forces the pharmaceutical industry to face major challenges. Thus, the present manuscript could be of interest for the pharmaceutical researches.

The Authors made a lot of experimental work and used a number of techniques.

1.1.- The manuscript is quite well written but requires a revision of the English language and some corrections (see, for example, pag. 2, line 66; pag 4, lines 161-162, 182; pag. 6, lines 248, 262; pag. 7 line 290 etc.).

Answer: The entire text was sent to be reviewed by a scientific English language expert so that the manuscript can be better understood

1.2.- With regard to the FT-IR measurements, the Authors should specify which is the composition of the physical mixture whose spectrum is reported in figure 3.

Answer: The composition of the physical mixture is added in a 1:3 molar ratio in figures 3 and 4. This has been included in the manuscript.

1.3.- Furthermore, the spectra of all the physical mixtures with the same composition of the complexes should be shown for a comparison with the spectra of the complexes.

Answer: Only the physical mixture (1:3) was prepared, because this proportion corresponds to the worst case scenario where the amount of API is in a lower proportion; therefore, the conclusions are based on the fact that the signal reduction is due to the crystalline change of the drug and does not depend on the dilution of it. The observed API signals do not smear in the physical mixture, which is independent of concentration, appearing at the same wave number.

1.4.- DSC measurements could give important quantitative information starting from the lurasidone melting peak. Can the Authors carry out DSC measurements?

Answer: The measurements requested for DSC are included in Figure 6, with their corresponding methods, results and discussion.

1.5.- Pag. 14, line 434-435: A compound in the amorphous state has higher solubility than in the crystalline phase. The Authors wrote the opposite. Please, correct.

Answer: Text is corrected as requested

1.6.- The paragraph of the Conclusions is poor. The Authors could merge part of the Discussion to the Conclusion paragraph.

Answer: Part of the discussion was incorporated to the conclusion to improve this section

Reviewer 2 Report

The article entitled "Improving lurasidone HCl solubility and stability by higher-order complex formation with hydroxypropyl-β-cyclodextrin" adresses an interesting topic, contains interesting results and discussion. However, to me, some aspects have to be adressed/changed regarding :

- abstract : nothing is mentionned about the issues of lurasidone stability, this should be added so as to understand why it is important to improve this aspect ; lines 33-36 should be rephrased. 

- introduction : some aspects are clearly lacking :

* lines 72 to 75 : the authors depict procedures to increase solubility, however the reasons that has driven the choice of formation of inclusion complexes is not explainde

* lines 102-107 : these lines, about the general context should be proposed earlier in the text

* The authors should also detail what is known about the chemical stability of lurasidone (LRD) as they investigate that point

- Regarding material and method and results :

- did the authors compare the dissolution results to what was obtained for tablets?

- regarding stability results :

* where the HPLC conditions verified so as to check that no co elution of degradation products of LRD occurs and also to check that the method has a sufficient detection limit to detect the degradation products of LRD ? Is the method stability indicating? This is necessary to check this aspect Why a detection at 317 nm?

*the authors mention a study on the stability of LRD (ref 41) : the results of this study should be more detailed so as to understand the propensity of LRD to degrade

*Furthermore, in ref 41 it was found that LRD is instable in the presence of hydrogen peroxide. However the authors did not assess if LRD is protected against this stress when using the formulation. The authors should comment on that

- the discussion is very long and the authors should enlighten it by discussing only the results. In that sense, I suggest to remove lines 467-491 or at least not to reintroduce data on Sporanox,as it has already been depicted in the introduction

Author Response

The article entitled "Improving lurasidone HCl solubility and stability by higher-order complex formation with hydroxypropyl-β-cyclodextrin" adresses an interesting topic, contains interesting results and discussion. However, to me, some aspects have to be adressed/changed regarding :

2.1.- Abstract: nothing is mentioned about the issues of lurasidone stability, this should be added so as to understand why it is important to improve this aspect ; lines 33-36 should be rephrased.

Answer: LRD stability issues are now included in the abstract

2.2.- Introduction: some aspects are clearly lacking:

* lines 72 to 75: the authors depict procedures to increase solubility, however the reasons that has driven the choice of formation of inclusion complexes is not explained

Answer: The explanation on the criteria for choosing the cyclodextrin inclusion complex is now detailed.

2.3.- Lines 102-107: these lines, about the general context should be proposed earlier in the text

Answer: We rearranged the text in the introduction as requested by the reviewer

2.4 The authors should also detail what is known about the chemical stability of lurasidone (LRD) as they investigate that point

Answer: The Introduction now details this aspect as requested by the reviewer

2.5.- Regarding material and method and results:

- did the authors compare the dissolution results to what was obtained for tablets?

Answer: The results are compared with capsules according to Londhe et al (2018).

2.6.- Regarding stability results:

* where the HPLC conditions verified so as to check that no co elution of degradation products of LRD occurs and also to check that the method has a sufficient detection limit to detect the degradation products of LRD? Is the method stability indicating?

Answer: To demonstrate that there is no coelution between LRD and its degradation products, the specificity evaluation was carried out (see table 5). The partial validation of the method was carried out according to ICH Q2 (see table 4), the linearity results indicate that the method is linear from the concentration of the working solution, which is 1 mg/mL, to the concentration of the detection limit 0.6 ug/mL, the detection limit chromatogram is added in Supplementary Figure S7. Due to the above, the detection of any impurity can be ensured, being the analytical method indicative of stability.

This is necessary to check this aspect Why a detection at 317 nm?

Answer: Explanation added in methodology 2.2: This wavelength was selected based on its absorption spectrum that presents a maximum at 317 nm associated with the benzothiazole chromophore (Figure S6).

2.7.- The authors mention a study on the stability of LRD (ref 41) : the results of this study should be more detailed so as to understand the propensity of LRD to degrade

Answer: The Introduction rearranges the text and presents this aspect in more detail now as requested by the reviewer

2.8.- Furthermore, in ref 41 it was found that LRD is instable in the presence of hydrogen peroxide. However the authors did not assess if LRD is protected against this stress when using the formulation. The authors should comment on that

Answer: The result of the stability evaluation under oxidative conditions is incorporated

2.9.- The discussion is very long and the authors should enlighten it by discussing only the results. In that sense, I suggest to remove lines 467-491 or at least not to reintroduce data on Sporanox, as it has already been depicted in the introduction

Answer: The text suggested by the reviewer was removed, and part of the discussion is now incorporated into the conclusion

Reviewer 3 Report

This is the work on the complexation of a drug with HP-BCD, and as such is pretty standard. Although authors apply many methods to support their findings it still lacks some significant data - e.g establishing stability constants for the obtained complex - it is known that applying cyclodextrins as drug delivery systems is only valid when the complexes are stable enough (authors could use any posiible method to measure it, e.g. 1H NMR, method that was mentioned in the manuscript). There is no relevant data to the previous research concentrating on luracidone HCl. Moreover, in the Introduction section, some relevant reviews, regading use of cuclodextrins as drug delivery systems should be cited. Finally, presented study regards only one new compound, using standard procedures. In my opinion it is not enough for publication

Author Response

This is the work on the complexation of a drug with HP-BCD, and as such is pretty standard. 3.1.-Although authors apply many methods to support their findings it still lacks some significant data - e.g establishing stability constants for the obtained complex - it is known that applying cyclodextrins as drug delivery systems is only valid when the complexes are stable enough (authors could use any posiible method to measure it, e.g. 1H NMR, method that was mentioned in the manuscript).

Answer: The requested stability constants are calculated through computational modeling

3.2.- There is no relevant data to the previous research concentrating on luracidone HCl.

Answer: Previous research on improving the solubility of LRD is incorporated into the introduction.

3.3.- Moreover, in the Introduction section, some relevant reviews, regarding use of cyclodextrins as drug delivery systems should be cited.

Answer: The Introduction was expanded to include this literature as requested by the reviewer

3.4.- Finally, presented study regards only one new compound, using standard procedures. In my opinion it is not enough for publication

While this is the study of only one compound, we believe that the experimental design, together now with computer simulation studies, and the interesting results found are of great relevance to the field and would be of interest to the broad readership of Pharmaceutics.

Round 2

Reviewer 1 Report

The Authors made the required revision to their manuscript, however, some changes are necessary. In particular:

-Section 2.7: 

Please, change the sentence …was placed in an aluminum pan and evaluated with a temperature gradient of 25 °C to 350 °C at 10 °C/min.

to: …was placed in an aluminum pan and heated at 10 °C/min from 25 °C to 350 °C. 

-Section 3.5:

Please, replace “thermogram” with “DSC curve” or “thermoanalytical curve”, according to the IUPAC Recommendations.

Furthermore, it is advisable to report the value of onset temperature of the peaks instead that of the maximum/minimum of the peak because this last strongly depends on the experimental conditions such as the sample mass. In some cases, you can report the temperature range of the peak. 

The shape of the peak shown in the DSC curve of LRD suggests that at those temperature the active principle is undergoing melting immediately followed by decomposition. Thus, please, change the sentence “LRD presented an endo-thermic signal associated with the melting point of the drug…” to “LRD presented an endo-esothermic effect associated with melting and decomposition of the drug.”

I suggest changing this paragraph: “In the physical mixture, the drug fusion signal was observed at a lower intensity and at a lower temperature (about 241°C), which was also reported in a physical mixture of BCD with LRD, where the signal was present at approximately 240°C [38]. In the 1:2 and 1:3 LRD-HPBCD ratios, thermograms did not have significant peaks indicating the amorphous nature of the solid. This was consistent with the X ray analysis of the Figure 5. The LRD-HPBCD (1:0.5) complex showed a significant endothermic signal at 233°C indicating the presence of crystalline drug material. This phenomenon occurred to a lesser extent in the LRD-HPBCD (1:1) complex”. 

to:

“In the physical mixture, the drug melting/decomposition were observed at a significantly lower temperature, in according to what already reported for a physical mixture of BCD with LRD [38]. In the 1:2 and 1:3 LRD:HPBCD ratios, the DSC curves did not show the endothermic peak indicating the amorphous nature of the solid. This was consistent with the X-ray analysis of the Figure 5. The LRD-HPBCD (1:0.5) complex showed a significant endothermic signal at the onset temperature … °C, indicating the presence of crystalline drug. This phenomenon occurred to a lesser extent in the LRD-HPBC (1:1) complex”.

Finally, I suggest: - to erase in Figure 6 the dot lines that indicate the integration of the peaks, - to add “arbitrary counts” in the Y axis title, - to change the X-axis title to “Temperature (°C)”.

Author Response

The Authors made the required revision to their manuscript, however, some changes are necessary. In particular:

-Section 2.7: Please, change the sentence …was placed in an aluminum pan and evaluated with a temperature gradient of 25 °C to 350 °C at 10 °C/min. to: …was placed in an aluminum pan and heated at 10 °C/min from 25 °C to 350 °C. 

Answer: The changes suggested by the reviewer were made to the text

-Section 3.5: Please, replace “thermogram” with “DSC curve” or “thermoanalytical curve”, according to the IUPAC Recommendations.

Answer: The figure N°6 was corrected based on reviewer suggestions and the text was changed.

Furthermore, it is advisable to report the value of onset temperature of the peaks instead that of the maximum/minimum of the peak because this last strongly depends on the experimental conditions such as the sample mass. In some cases, you can report the temperature range of the peak. 

Answer: Table 2 is now included and the ranges recommended by the reviewer are indicated.

The shape of the peak shown in the DSC curve of LRD suggests that at those temperature the active principle is undergoing melting immediately followed by decomposition. Thus, please, change the sentence “LRD presented an endo-thermic signal associated with the melting point of the drug…” to “LRD presented an endo-esothermic effect associated with melting and decomposition of the drug.”

Answer: The changes suggested by the reviewer were made to the text

I suggest changing this paragraph: “In the physical mixture, the drug fusion signal was observed at a lower intensity and at a lower temperature (about 241°C), which was also reported in a physical mixture of BCD with LRD, where the signal was present at approximately 240°C [38]. In the 1:2 and 1:3 LRD-HPBCD ratios, thermograms did not have significant peaks indicating the amorphous nature of the solid. This was consistent with the X ray analysis of the Figure 5. The LRD-HPBCD (1:0.5) complex showed a significant endothermic signal at 233°C indicating the presence of crystalline drug material. This phenomenon occurred to a lesser extent in the LRD-HPBCD (1:1) complex”. 

to: “In the physical mixture, the drug melting/decomposition were observed at a significantly lower temperature, in according to what already reported for a physical mixture of BCD with LRD [38]. In the 1:2 and 1:3 LRD:HPBCD ratios, the DSC curves did not show the endothermic peak indicating the amorphous nature of the solid. This was consistent with the X-ray analysis of the Figure 5. The LRD-HPBCD (1:0.5) complex showed a significant endothermic signal at the onset temperature … °C, indicating the presence of crystalline drug. This phenomenon occurred to a lesser extent in the LRD-HPBC (1:1) complex”.

Answer: The changes suggested by the reviewer were made to the text

Finally, I suggest: - to erase in Figure 6 the dot lines that indicate the integration of the peaks, - to add “arbitrary counts” in the Y axis title, - to change the X-axis title to “Temperature (°C)”.

Answer: Figure 6 was corrected based on reviewer suggestions

Reviewer 2 Report

The authors have responded to the issues I previously raised in the review. Furthermore, very interesting theory-based data (docking) have been added, supporting the formation of complexes.

To me, the manuscript is appropriate for publication.

Author Response

The authors have responded to the issues I previously raised in the review. Furthermore, very interesting theory-based data (docking) have been added, supporting the formation of complexes. To me, the manuscript is appropriate for publication.

Answer: The entire text was sent to be reviewed by a scientific English language expert so that the manuscript can be better understood